# TGF-β-Mediated Epithelial-Mesenchymal Transition and Cancer Metastasis

**DOI:** 10.3390/ijms20112767

**Published:** 2019-06-05

**Authors:** Yang Hao, David Baker, Peter ten Dijke

**Affiliations:** Department of Cell and Chemical Biology and Oncode Institute, Leiden University Medical Center, Einthovenweg 20, 2300 RC Leiden, The Netherlands; Y.Hao@lumc.nl (Y.H.); D.A.Baker@lumc.nl (D.B.)

**Keywords:** EMT, lncRNA, metastasis, miRNA, SMAD, TGF-β, targeted therapy, tumor microenvironment

## Abstract

Transforming growth factor β (TGF-β) is a secreted cytokine that regulates cell proliferation, migration, and the differentiation of a plethora of different cell types. Consistent with these findings, TGF-β plays a key role in controlling embryogenic development, inflammation, and tissue repair, as well as in maintaining adult tissue homeostasis. TGF-β elicits a broad range of context-dependent cellular responses, and consequently, alterations in TGF-β signaling have been implicated in many diseases, including cancer. During the early stages of tumorigenesis, TGF-β acts as a tumor suppressor by inducing cytostasis and the apoptosis of normal and premalignant cells. However, at later stages, when cancer cells have acquired oncogenic mutations and/or have lost tumor suppressor gene function, cells are resistant to TGF-β-induced growth arrest, and TGF-β functions as a tumor promotor by stimulating tumor cells to undergo the so-called epithelial-mesenchymal transition (EMT). The latter leads to metastasis and chemotherapy resistance. TGF-β further supports cancer growth and progression by activating tumor angiogenesis and cancer-associated fibroblasts and enabling the tumor to evade inhibitory immune responses. In this review, we will consider the role of TGF-β signaling in cell cycle arrest, apoptosis, EMT and cancer cell metastasis. In particular, we will highlight recent insights into the multistep and dynamically controlled process of TGF-β-induced EMT and the functions of miRNAs and long noncoding RNAs in this process. Finally, we will discuss how these new mechanistic insights might be exploited to develop novel therapeutic interventions.

## 1. Introduction

Cancer treatments have been refined over a period spanning several thousand years, culminating in the primary modern approaches of surgery, chemotherapy and radiation therapy [1]. In recent decades, more targeted and personalized treatments have gained prominence. The fundamental aim of these therapies is to specifically kill tumor cells while leaving healthy tissues intact [2]. Despite demonstrable progress, these strategies frequently deliver relatively modest improvements in disease outcomes owing to acquired resistance and excessive toxicity [3]. These findings indicate the need for greater optimization of current treatments as well as the identification of alternative targets for the development of novel cancer remedies.

Cancer is a complex disease in which tumor cell heterogeneity and the reciprocal interplay between tumor cells and surrounding stromal cells and extracellular matrix are key determinants in tumor progression and therapy response. Tumorigenesis shows similarity to subverted normal embryogenic developmental processes in which communication between cells is controlled by cytokines that act in an autocrine, paracrine or juxtacrine manner. In this light, we will provide a general overview of the established roles of one such important developmental cancer signaling pathway, namely the transforming growth factor-β (TGF-β), in tumorigenesis [4]. In particular, we consider how this essential TGF-β signaling network orchestrates the epithelial to mesenchymal transition (EMT), the mechanism by which cancer cells lose polarity and separate from each other, adopt the characteristics of a mesenchymal phenotype, become motile and invade distant sites. Key TGF-β-induced effectors in this process are the transcriptional repressors of E-cadherin, e.g., SNAIL1, SNAIL2 (also termed SLUG), ZEB1/2 and TWIST. Moreover, miRNAs and long noncoding RNAs (lncRNAs) are emerging as potentially quantifiable biomarkers of cancer status and are potential targets for anti-metastatic therapies [5]. In this review, we will highlight the pivotal role of these molecules in regulating TGF-β signaling and epithelial-mesenchymal transition (EMT).

## 2. TGF-β and Signaling Transduction across the Plasma Membrane

TGF-β1 (hereafter termed simply TGF-β) is the prototypic member of a large family of structurally and functionally related proteins, which includes its close relatives TGF-β2 and TGF-β3 but also activins, nodal, inhibins, Mullerian-inhibiting substance (MIS), growth and differentiation factors (GDF) and bone morphogenetic proteins (BMPs) [6]. TGF-β was discovered in the early 1980s as a secreted factor that, together with TGF-β or epidermal growth factor (EGF), induced the growth of normal rat kidney (NRK) cells in soft agar. Since then, we have gained a much deeper understanding of this multifunctional cytokine [7]. The TGF-β gene encodes a pre-pro-precursor peptide of 390 amino acids. The pro-precursor peptide is proteolytically processed by furin proteases into an amino-terminal fragment and the carboxy-terminal 112 amino acids, which corresponds to the mature bioactive TGF-β [8]. The amino-terminal part is also termed the latency associated peptide (LAP) and is noncovalently attached to the mature TGF-β [9,10]. Latent TGF-β is activated by specific proteases that cleave the LAP and/or by mechanical forces generated by cell surface integrins, resulting in the release of mature, active TGF-β [11,12]. Bioactive TGF-β, which is capable of binding its cell surface receptors, is a dimeric protein linked by disulfide bonds with an apparent molecular weight of 25 kDa.

TGF-β exerts its cellular effects via cell surface TGF-β type I and type II receptors, e.g., TβRI and TβRII, respectively [13]. TβRI and TβRII are structurally related and consist of an extracellular domain characterized by the presence of cysteine residues that form disulfide bonds, a single transmembrane domain and an intracellular region harboring a conserved serine/threonine kinase domain. TGF-β initially engages the TβRII, which drives the recruitment of TβRI and the formation of a heterotetrameric complex composed of two TβRIIs and two TβRIs. Subsequently, the active TβRII kinase domain phosphorylates TβRI on specific serine and threonine residues in the glycine-serine (GS) juxtamembrane region, which leads to its activation [14]. TGF-β receptors are widely expressed in human tissues/cells and mediate various biological phenomena, such as embryonic development, tissue homeostasis, organogenesis, immune surveillance and tissue repair.

## 3. Intracellular SMAD and Non-SMAD Signaling Pathways

Genetic studies designed to delineate the critical components of the dauer and decapentaplegic pathway in *Caenorhabditis elegans* and *Drosophila*, respectively, led to the identification of *Small* (*Sma*) and *Mothers against dpp* (*Mad*) genes [15,16]. The mammalian homologues of the proteins encoded by these genes, termed SMADs, were found to act as intracellular transcriptional effectors of TGF-β family receptor signaling [17]. The SMAD family can be divided into receptor-regulated (R-) SMADs (in vertebrates: R-SMAD1, -2, -3, -5 and -8) that interact with and become phosphorylated by activated type I receptor kinases, the common (Co-) SMADs (in vertebrates: SMAD4) that form heteromeric complexes with activated R-SMADs and inhibitory I-SMADs (in vertebrates: I-SMAD6/7), which antagonize canonical SMAD signaling [18] (Figure 1).

The R- and Co-SMADs have two conserved domains, termed Mad homology (MH)1 and MH2 domains at the amino-terminal (MH1) and carboxy-terminal (MH2) ends of the proteins. MH1 and MH2 are separated by a flexible linker region. R-SMAD phosphorylation by type I receptors occurs on two serine residues, which comprise the SXS sequence motif, at the C-termini. SMAD3 and SMAD4, but not SMAD2, bind directly to the consensus 5′-CAGA-3′ DNA motif. Heteromeric complex formation between R-SMADs and SMAD4 is mediated by MH2 domains. Heteromeric complex formation of R-SMADs and SMAD4 exposes nuclear import signals and shields nuclear export signals, resulting in their nuclear accumulation. In the nucleus, they can act as transcription factors in concert with other DNA binding transcription factors, coactivators and repressors and chromatin remodeling factors, which enable diverse transcriptional responses depending upon the particular combination of proteins [19]. I-SMAD7, which has a carboxy terminal region with homology to R-SMADs and SMAD4 MH2 domains, antagonizes the activation of TGF-β receptor/SMAD signaling. SMAD7 achieves this via multiple mechanisms, including the recruitment of the E3 ubiquitin ligase SMURF2 to the activated receptor, thereby targeting the TGF-β receptor for proteasomal and lysosomal degradation. SMAD7 can also attenuate signaling by recruiting phosphatases to the activated TGF-β receptor, which mediate receptor inactivation by the dephosphorylation of specific amino acid residues [20].

TGF-β can also signal via non-SMAD pathways. This process can occur directly or indirectly [21]. An example of an indirect mechanism is TGF-β/SMAD-stimulated expression of growth factors, such as platelet-derived growth factor (PDGF) and epidermal growth factor (EGF), which thereafter initiate non-SMAD responses. Non-SMAD signaling can also be initiated directly downstream of the TGF-β receptor by activation of various branches of the MAP kinase (MAPK) pathway, Rho-like GTPase signaling pathways and the phosphatidylinositol-3-kinase (PI3K)/AKT pathway [22]. For example, activated TβRI can induce tyrosine phosphorylation of SHCA, which associates with GRB2, and recruits a GRB2/SOS complex, thereby triggering activation of the RAS/RAF/MAP kinase signaling cascade [23]. Activated TGF-β receptor complexes also induce K63-linked polyubiquitination of TRAF4/6, leading to the recruitment of TAK1 and triggering its activation, thus allowing TAK1 to activate JNK signaling through MKK4 or P38 MAPK signaling via MKK3/6 [18,24,25]. At tight junctions, TβRII phosphorylates PAR6 at serine residue 345. This phosphorylated PAR6 recruits SMURF1 to the activated TβRI-TβRII receptor complex. The PAR6-SMURF1 complex subsequently mediates localized ubiquitination and turnover of RHOA at cellular protrusions [26]. TGF-β can promote the interaction between CDC42 GTPase/RAC and p21-activated kinase (PAK) 2, an interaction that is dependent on SMAD7 [27]. TGF-β receptors interact with the regulatory p85 subunit of PI3K, resulting in activation of the PI3K/AKT pathway, which controls translational responses through mammalian target of rapamycin (mTOR) [28]. Activation of non-SMAD signaling occurs in a context-dependent manner, and these pathways also crosstalk with the canonical SMAD pathway.

## 4. Tumor Suppressive Effects of TGF-β

TGF-β can act as a potent tumor suppressor in normal and premalignant epithelial cell types (Figure 2). TGF-β triggers G1 phase cell cycle arrest by different mechanisms in different cell types [29]. For example, this molecule can activate the translation-inhibitory protein 4E-BP1 (regulator of eukaryotic translation initiation factor-4F (eIF4E)) promoter activity through SMAD4, thereby suppressing translation and cell growth and proliferation [30]. TGF-β causes late G1 cell cycle arrest by inducing the expression of cyclin-dependent kinase (CDK) inhibitors (p15INK4b, p21WAF1 and p27KIP1) to inhibit CDK-cyclin complexes [31]. In MCF10A and MDA-MB-231 cell lines, the tumor suppressor p53 is a critical SMAD partner, which promotes TGF-β-induced p21 expression to block cell cycle progression [32]. In addition, TGF-β exerts growth inhibitory effects by inhibiting the expression of CDC25a phosphatase (which is required for CDK-cyclin activation) [33] or negatively regulating Id proteins (helix-loop-helix (HLH) proteins, which are essential for inhibition of cell differentiation and growth arrest) [34] in the prostate epithelial cell line HPr-1. TGF-β directly inhibits c-MYC by binding a transcriptional repression complex containing SMAD2/3, E2F4/5, p107, and C/EBPβ to the TGF-β inhibitory element in the proximal region and thus achieving cell cycle arrest in HaCaT, COS-1, and Mv1Lu-tet-p15 cells and human leukemia MO-91 cells [35].

In addition to cell cycle arrest, TGF-β can induce cell apoptosis in the early phase of tumorigenesis [29]. However, the precise mechanisms by which TGF-β induces this effect in different cell types remain unclear. The expression of some apoptotic regulators (such as growth arrest and DNA damage (GADD) 45, Bcl-2-like protein 11 (BIM), BCL-2 interacting killer (BIK), death associated protein kinase (DAPK), FAS, and B-cell lymphoma-extra large (BCL-XL)) was shown to be regulated by the TGF-β/SMAD signaling pathway [36]. For example, BIM was found to be a key mediator of TGF-β-induced apoptosis in intestinal adenoma cells [37] and in hepatocarcinoma cells [38]. TGF-β1-associated regulatory SMAD proteins bind to the BIK (also known as NBK) promoter, which encodes a proapoptotic sensitizer protein in B cells [39]. After TGF-β1 treatment, researchers found that TGF-β induced SMAD-dependent binding between the proapoptotic effector BIM and BCL-XL in gastric carcinoma cell lines [40] and a decrease in BCL-XL expression followed by activation of the apoptosis proteins caspase-9 and caspase-3 in human hepatoma cells (HuH-7) [41]. TGF-β can activate the TAK1-p38/JNK pathway, which has been reported to lead to apoptosis in HEK 293T cells [25]. This molecule also promoted the expression of SMAD-dependent GADD45β in hepatocytes and plays an important role in cell death by mediating delayed TGF-β-induced p38 MAP kinase activation [42]. Here, the effects of TGF-β in proapoptotic signaling occur in a context-dependent manner. Additionally, the TGF-β signaling pathway can be coupled to the cell death machinery through the induction of reactive oxygen species (ROS) [43], apoptosis genes (SHIP and TIEG), modulation of epigenetic regulators (DNMTs) [44], H3K79me3 and H2BK120me1 [45], and telomere shortening through regulating human telomerase reverse transcriptase (hTERT) in the breast cancer MCF-7 cell line [46]. These effects of TGF-β in proapoptotic signaling occur in a context-dependent manner.

The TGF-β signaling pathway can inhibit tumor growth by multiple other mechanisms, including activating autophagy in certain human cancer cells. For example, in human hepatocellular carcinoma cell lines, TGF-β induced the accumulation of autophagosomes and increased the expression levels of the autophagy markers Autophagy-related 5 (ATG5), Beclin1, ATG7 and death-associated protein kinase (DAPK) [47]. Moreover, siRNA-mediated silencing of autophagy genes attenuated TGF-β-mediated growth inhibition and induction of the proapoptotic genes BIM and BMF in human hepatocellular carcinoma cells [48].

Because of its central role in tumor suppression, TGF-β signaling components were found to be mutated and functionally inactivated in various cancers [49]. The first example was SMAD4, which is frequently mutated in gastrointestinal cancers [50]. Subsequently, other TGF-β signaling components, e.g., TGF-β receptors [51] and SMADs (SMAD2 and SMAD3), were found to be mutated in various cancers, including bladder, colon, breast, esophageal, stomach, brain, liver, and lung cancers [52]. Loss of tumor suppressor function and epigenome and microenvironmental changes can also affect the tumor-promoting activity of the TGF-β receptor/SMAD pathway [53]. For example, in gastrointestinal tumors, TβR1 activity was decreased due to the methylation status of the *TβR1* promoter [54]. In turn, TGF-β/SMAD can affect the epigenome of genes involved in cancer processes. TGF-β and SMAD2/3 show oncogenic activities, such as promoting glioma cell proliferation, by affecting the methylation status of the *platelet-derived growth factor-β* (*PDGF-B*) gene and autocrine PDGF-B signaling within tumor microenvironments [55]. TGF-β stimulated myofibroblast percent and invasion rate in tumor-associated fibroblasts (CAFs) that increase tumor invasion [56].

## 5. TGF-β-induced Tumor Promoting Effects

### 5.1. Cell Biology of TGF-β-induced EMT

In the late stage of cancer progression, cancer cells remain responsive to TGF-β but become resistant to its cytostatic effects. In fact, by acting directly on cancer cells, TGF-β can promote tumorigenesis by inducing the so-called epithelial to mesenchymal transition (EMT) [57]. Under normal physiological conditions, EMT plays a crucial role in the context of embryogenesis and tissue damage repair [58]. This process can be subverted and pathological, and EMT drives the development of fibrotic disease and tumorigenesis [59]. EMT is characterized by changes in the levels of three prominent biomarkers (E-cadherin, vimentin, and N-cadherin), and these changes can lead to decreased adhesion of cells, loss of polarity and tight junctions. At the same time, epithelial cells adopt the traits of a mesenchymal phenotype, notably motility and susceptibility to invasion and metastasis (Figure 3) [60]. Importantly, mesenchymal cancer cells are correlated with poor prognosis and associated with resistance to chemotherapy [61]. Interestingly, recent studies have questioned the necessity of EMT in establishing metastasis [62]. Zheng et al. reported that in genetically engineered mouse models of pancreatic adenocarcinoma development and spontaneous metastasis mouse models of breast cancer, tumor cells could metastasize without activating EMT programs, and EMT only contributed to chemoresistance [63]. Similarly, Fischer et al. also showed that EMT is not required for lung metastasis but contributes to chemoresistance in spontaneous breast-to-lung metastasis models [64]. However, these findings have been challenged by other researchers, who believed that Zheng et al. failed to completely suppress the activation of EMT, and their results can only speak to the redundancy of EMT within the transcriptional network in pancreatic carcinomas. These researchers continue to subscribe to the notion that EMT is required for metastatic dissemination in pancreatic carcinoma cells [65].

Increasing data have shown that EMT is not a single, stereotypical program but instead has a multistep process that passes through intermediate hybrid states (partial EMT state, P-EMT) during the transition from epithelial to mesenchymal cells [66]. The TGF-β-induced transition from epithelial to P-EMT is reversible, whereas the transition from P-EMT to mesenchymal cells is potentially irreversible depending on the type of cell that is involved [59]. Studies have found that there are at least seven tumor subpopulations associated with different EMT stages in skin and breast cancer tissues, which contributes to intratumor heterogeneity. These EMT subpopulations displayed differences in cellular plasticity, invasiveness, and metastatic potential, and tumor cells with an early stage of EMT were most likely to metastasize [67]. An interesting paper reported that TGF-β activates scleroderma epithelial cells to the P-EMT process in fibrotic skin [68]. Related to this, single cell RNA-seq showed that P-EMT plays an important role in head and neck cancer, and further in vitro analyses suggested that TGF-β dynamically controls the transition between P-EMT and non-P-EMT states in cells [69].

### 5.2. Molecular Mechanisms in TGF-β-induced EMT

SMAD levels/activities mediate TGF-β-induced EMT by inducing the expression of E-cadherin transcriptional repressors, such as SNAIL, ZEB and TWIST, which cooperate with other transcription regulators in the nucleus [58]. Several additional lines of evidence argue for a role of SMADs in TGF-β-induced EMT, and some examples are provided below. Knockdown of SMAD4 in MDA-MB-231 breast cancer cells robustly attenuated bone metastasis in nude mice and significantly prolonged survival of the treated animals [70]. SMAD3 and SMAD4 interact to form a complex with SNAIL1 that targets the tight-junction protein (CAR) and E-cadherin during TGF-β-driven EMT in breast epithelial cells. Conversely, co-silencing of SNAIL1 and SMAD4 by siRNA inhibited repression of CAR and occludin during EMT [71]. In addition, SMAD3/SMAD4-mediated SNAIL transcription contributed to EMT during skin carcinogenesis, while SMAD2 loss significantly increased this effect [72]. Moreover, SMAD7, the transcriptional target and negative regulator of TGF-β signaling, upregulated TGF-β and inducing SMAD7 transcription prevented TGF-β-induced EMT and invasion of cancer cells [73]. Additionally, ubiquitin ligases that promote poly-ubiquitination and proteasomal degradation of SMADs affect EMT. This finding is illustrated by E3 ubiquitin ligase RNF8, which activates TWIST via K63-linked ubiquitination to promote EMT and cancer stem cell (CSC) self-renewal, resulting in enhanced metastasis and chemoresistance in breast cancer [74].

TGF-β can also induce EMT in a non-SMAD-dependent fashion, for example, by promoting cytoskeletal remodeling, which leads to activation of ERK [75]. The ERK required for cytoskeletal remodeling interacts with SHC or GRB2 to form an SHC-GRB2-ERK complex, which is a key component of TGF-β-induced tumor invasion and metastasis [76]. ERK substrates, AP-1 family members, enhance SMAD transcriptional activity to regulate gene expression and TGF-β-induced EMT [77]. However, the RHO-like GTPases, including RHOA, RAC and CDC42, are also involved in TGF-β-induced EMT. TGF-β regulates cytoskeletal changes via mediating RHO GTPase to achieve the dissolution of tight junctions among cells. TGF-β mediates the RHOA activity level and promotes the activation of LIM kinase (LIMK) by Rho-related kinase (ROCK) and phosphorylated myosin light chain (MLC) to inhibit cofilin [78]. In addition, TGF-β affects tight junctions through SMAD7-dependent CDC42-PAK1 (p21-activated kinase) and filopodia formation. The TRAF6-TAK1-JNK/P38 pathway and PI3K-AKT-mTOR signaling are also necessary non-SMAD pathways for TGF-β-mediated EMT [79,80]. Scientists have shown that PI3K/AKT signaling promotes tumor metastasis by inducing TWIST1 phosphorylation, via a crosstalk between AKT/PKB and TGF-β signaling [81]. Twist also had a significant effect on AKT signaling pathway activation by inducing expression of miR-10b in gastric cancer cells, and the miR-10b induced by TWIST increased the expression of a well-characterized pro-metastatic gene, RHOC [82]. Moreover, in an orthotopic syngeneic mouse tumor model, metastasis caused by EMT was attenuated in mice treated with the p38 inhibitor SB203580 [83]. TRAF6 knockdown inhibited the migration and invasion caused by EMT of SCCHN (squamous cell carcinoma of head and neck) cells [84].

In addition to these SMAD/non-SMAD pathways, TGF-β affects the activities of other EMT trigger signaling pathways (NOTCH, WNT, INTEGRIN, etc.) by several complexes, such as ZEB1/2, the SNAIL1-SMAD3/4 complex, the β-catenin-SMAD2 complex, the LEF1-SMAD3/4 complex and the SMAD3-AP1-1 complex [85]. As early as 20 years ago, scientists discovered that SMAD4 could form a complex with β-catenin and LEF1/TCF (lymphoid enhancer factor1), which are downstream components of the Wnt signaling cascade in vivo [86]. SMAD signaling subsequently stimulates the formation of β-catenin/LEF1 and SNAIL-LEF1 complexes, which promote EMT by inhibiting the expression of E-cadherin [87,88]. TGF-β can promote EMT associated with WNT-11 signals through the WNT-11 receptor FZD8 in prostate cancer [89]. Moreover, WNT-11 signaling mediates the nuclear entry process of TAK1 (TGF-β-activated kinase) [90]. The TGF-β and NOTCH pathways coregulate a large cohort of genes in human cancer, such as renal cell carcinoma [91]. R-SMAD activates the NOTCH ligand Jagged1 to release Notch intracellular domain (ICN) and then binds to CLS (an acronym for *C*BF-1/RBPJ-κ in *Homo sapiens*/*Mus musculus* respectively, *S*uppressor of Hairless in *Drosophila melanogaster*, *L*ag-1 in *Caenorhabditis elegans)*. This ICN-CLS complex induces the binding of the transcription factor SNAIL or HEYl to the E-cadherin E-box to reduce E-cadherin expression and initiate the EMT process [92]. Moreover, SMAD signaling and MAPK/JNK signaling converge at AP1-binding promoter sites by SMAD3 and SMAD4, which cooperate with c-JUN/c-FOS [93], and the RAS-ERK MAP kinase pathways are likely to act synergistically with TGF-β and contribute to multiple aspects of the EMT, including the pro-invasive and pro-metastatic behavior of tumor cells of diverse tissue origins [94]. TGF-β increases the level of SNAIL and promotes EMT with the cooperation of oncogenic RAS [57] and the transcription factor nuclear factor κB (NF-κB) [95]. In addition, TGF-β upregulates receptors and ligands of PDGF, leading to phosphorylation of PI3K and activation of the SRC/STAT3 pathway, thereby triggering the EMT process [96].

### 5.3. MicroRNAs Involved in TGF-β-induced EMT

Two microRNA (a class of small noncoding RNAs approximately 22 nt in length)-dependent negative feedback loops are at the heart TGF-β-induced EMT (Figure 4). These pathways are the SNAIL1/miR-34 family/ZEB/miR-200 family feedback loop and the autocrine TGF-β/miR-200 feedback loop [97].

Mechanistically, TGF-β downregulates miR-200 family members, including miR-200a/-200b/-200c/-141/-429, which augments ZEB1 and ZEB2 mRNA levels. ZEB counteracts this mechanism through binding to the promoters of the miR-200 members and thereby repressing their expression. Additionally, miR-200 family members maintain the epithelial phenotype not only by targeting ZEB1/2 but also by actively repressing genes involved in cell motility and invasion [98]. MiR-1199-5p similarly regulates ZEB1 expression [99]. A comparable mechanism governs SNAIL1/miR-34 and the control of p53 status [100]. One study showed that in colorectal cancer, Zinc Finger protein 281 (ZNF281) can be an intermediate regulator between SNAIL1 and miR-34 [101]. In addition to SNAIL and p53, miR-34b experiences epigenetic regulation (chromatin modifications and DNA methylation) by directly targeting methyltransferases and deacetylases, resulting in a positive feedback loop inducing partial demethylation and activity [102]. Silencing miR-34a promoted liver metastases of colon cancer associated with upregulation of c-MET, SNAIL, and β-catenin expression [103]. Transcriptome profiling studies have demonstrated that TGF-β signaling regulates the SMAD4/miR-34a signaling network [104]. The SNAIL1/miR-34 regulatory loop was shown to be involved in the early reversible stage of EMT (from epithelial to P-EMT), whereas the ZEB/miR-200 system is responsible for the establishment of a mesenchymal state [105]. For the autocrine TGF-β/miR-200 system, autocrine TGF-β positively regulates the expression of SNAIL1 and then increases ZEB mRNA and protein levels, further affecting miR-200 [106]. This process makes the second switch (from P-EMT to mesenchymal) irreversible, modulating the maintenance of EMT.

High mobility group protein A2 (HMGA2) has been shown to promote lung cancer progression in mouse and human cells by competing with TGF-β type III receptor for the let-7 microRNA (miRNA) family [107], while decreased let-7g levels influence the TGF-β pathway by targeting TβR1 and SMAD2 gene expression [108]. The overexpression of miR-10b induced TGF-β-driven EMT in breast cancer [109]. In contrast, silencing of miR-10b markedly suppressed the formation of lung metastases by inhibiting its target gene HOXD10 in a mouse mammary tumor model [110].

TGF-β upregulates certain miRNAs, such as miR-182, which prolong NF-κB activation by directly suppressing an NF-κB negative regulator (cylindromatosis, CYLD) [111]. Overexpression of miR-182 restrained SMAD7 expression and promoted breast tumor invasion and TGF-β-induced osteoclastogenesis and bone metastasis [73]. MiR-181a, another miRNA upregulated by TGF-β, promoted TGF-β-mediated EMT and metastasis in breast cancer [112] and via repression of SMAD7 in ovarian cancer progression [113]. TGF-β activates miR-1269 by SOX4 and thereby enhances TGF-β signaling by targeting SMAD7 and HOXD10. This positive feedback loop significantly increased the ability of colorectal cancer cells to invade and metastasize in vivo [114]. Overexpression of miR-216a/217 activated the PI3K/AKT and TGF-β pathways by targeting PTEN, and SMAD7 underlies hepatocarcinogenesis and tumor recurrence of hepatocarcinoma [115]. TGF-β induces the expression and promoter activity of miR-155 through SMAD4. This change reduces RHOA protein and disrupts tight junction formation, leading to EMT [116]. Other research has shown that miR-206 inhibits autocrine production of TGF-β as well as downstream neuropilin-1 (NRP1) and SMAD2 expression, resulting in decreased migration, invasion, and EMT in breast cancer cells [117]. Several other miRNAs, such as miR-373, miR-655, miR206, miR-155, miR-140-5p, miR-494, miR-125a/b, and miR-375, have been implicated in EMT [118,119,120,121,122]. However, for many of these factors, their specific functions remain to be elucidated.

### 5.4. LncRNAs Involved in TGF-β-induced EMT

Long non coding RNAs (LncRNAs) are a class of RNAs that do not encode proteins or have minimal coding capacity. LncRNAs are emerging as important regulators of a variety of cellular and physiologic functions, such as chromatin dynamics, gene expression, growth, differentiation, and development [123]. LncRNAs can be differentially expressed and localized within the cell. They play a role in chromatin and DNA interactions, negatively or positively affecting the stability or processing of coding mRNA, directly binding to and modulating the functions of signaling proteins, and competitively binding to and thereby controlling the function of miRNAs [124]. Aberrant expression and mutations in lncRNAs have been linked to tumorigenesis, metastasis, and tumor stage [125]. Moreover, they have been detected in the circulating blood and/or urine of cancer patients. For example, plasma levels of a novel lncRNA, p53-induced transcript (Linc-pint), were significantly lower in patients with pancreatic ductal adenocarcinoma (PDAC) than healthy controls [126]. The expression of twenty lncRNAs linked with breast cancer-associated genes (BCAGs) was detectable in human breast cancer cell lines with different expression patterns [127]. LncRNAs are novel, potential therapeutic targets and biomarkers for cancer treatment, and new functions continue to be discovered [128].

Whereas previous studies have shown the involvement of miRNAs in regulating TGF-β signaling and EMT, only a few studies have reported a prominent role of lncRNAs in these processes. Trans-acting lncRNA ELIT-1 induced by TGF-β1 forms a positive TGF-β/SMAD3 signaling feedback and promotes EMT progression by acting as a SMAD3 cofactor [129]. The miR-17~92 polycistronic miRNA cluster encoded by the lncRNA MIR17HG locus was shown to attenuate the TGF-β signaling pathway and stimulate angiogenesis and tumor growth [65,130]. In a recent study, researchers found that a lncRNA activated by TGF-β, lncRNA-ATB, induces EMT and invasion by competitively binding miR-200 family members, which promoted organ-specific metastasis by binding IL-11 mRNA. This competitive binding increased IL-11 mRNA stability, which caused autocrine induction of IL-11 and subsequent activation of STAT3 signaling [131]. These findings suggest that lncRNA-ATB, a mediator of TGF-β signaling, could predispose HCC patients to metastasis [132]. Another study showed that lncRNA-PNUTS, which is highly expressed in mesenchymal breast tumor cells, competitive binds to and neutralizes the activity of miR-205 during EMT. Moreover, elevated expression of lncRNA-PNUTS was correlated with upregulated levels of ZEB mRNAs [133]. LncRNA MEG3 can modulate the activity of TGF-β genes by binding to distal regulatory elements [134]. Another lncRNA, DNM3OS, was associated with overexpression of TWIST1 and specifically contributed to EMT in ovarian cancer [135]. Recently, a paper showed that lncRNA-MUF can directly activate WNT/β-catenin signaling and EMT by binding to ANNEXIN 2A. LncRNA-MUF can also indirectly promote EMT by competitively binding to miR-34a and upregulating SNAIL1 expression [136].

Reduced lncRNA H19 expression in hepatocarcinogenesis (HCG) tissues from patients with the epithelial TGF-β gene signature [137] but increased H19 expression promoted tumor metastasis after TGF-β treatment in Hep3B cells [138]. In a mouse model of spontaneous metastatic breast cancer, lncRNA H19 mediated EMT and MET by differentially binding to the microRNAs miR-200b/c and let-7b [139]. LncRNA H19 can also interact with SLUG and/or EZH2, which regulates E-cadherin expression [140]. Lnc-Spry1 is downregulated by TGF-β and plays a direct regulatory role in the early stage of TGF-β-induced EMT, thus affecting cell invasion and migration. This molecule also controls gene and protein expression levels through an interaction with the splicing factor U2AF65 [141]. LncRNA-KRTAP5-AS1 and lncRNA-TUBB2A control the function of CLAUDIN-4 and thereby influence EMT in gastric cancer [142]. LncRNA HOTAIR (for HOX Transcript antisense intergenic RNA) acts as a crucial player during EMT by mediating a physical interaction between SNAIL and EZH2, which form an enzymatic subunit of the polycomb repressive complex 2, the main writer of chromatin-repressive marks [143]. Together, these findings suggest that lncRNAs can be mediators of TGF-β signaling and may serve as a potential target for anti-metastatic therapies. The mechanisms by which lncRNAs regulate TGF-β signaling are largely unknown, and how they affect TGF-β pathway components in cancer metastasis remains to be discovered (Figure 5 and Table 1).

## 6. TGF-β and Metastasis

### 6.1. TGF-β-induced Metastasis in Tissues

Tumor metastasis is the primary cause of cancer lethality; it is a progressive, multifactorial and multistep dynamic process, including detachment from a primary tumor, invasion into surrounding tissues, invasion into blood circulation/lymphatic circulation, survival in the circulatory system, extravasation from blood vessels, distal colonization, etc. [210]. The migration of cancer cells is pivotal to early metastasis, and changes in tumor cells (acquired by EMT) and the microenvironment are the two main factors that help cancer stem cells (CSCs) to escape from the primary site [211]. In the tumor microenvironment, TGF-β, Chemokine 4/12 (CXCL4/12), interleukin-6 (IL-6) and tumor necrosis factor-α (TNF-α), etc. can enhance EMT. At the same time, tumor cells secrete more epithelial growth factors, fibroblast growth factor (FGF) and insulin-like growth factor (IGF), leading to a hypoxic, acidic, high interstitial fluid pressure (IFP) state in the microenvironment, which activates cancer associated fibroblasts (CAFs) to produce more matrix metalloproteinases (MMPs) and remodel the tumor extracellular matrices (ECM) [212].

TGF-β induces metastasis to bone, liver, lung and other tissues of specific cancer types, such as breast, lung, gastric and prostate cancers (Figure 6A) [213]. Metastatic cancer cells have been shown to disturb the tight balance of bone transformation by osteoblasts and osteoclasts, conferring a receptive outgrowth microenvironment [214]. Tumor cells secrete cytokines and parathyroid hormone-related protein (PTHrP), which is the main inducer of osteoclast formation (also interleukin (IL)-1/-6/-11, etc.), and its expression is specific to the bone metastasis microenvironment [215]. TGF-β released in the active form upon osteoclastic bone resorption enhances PTHrP signaling in osteoblasts, resulting in osteoblasts expressing receptor activator of NFκB ligand (RANKL) while reducing osteoprotegerin (OPG) expression [216]. The high RANKL/OPG ratio enhances the osteolytic activity, which is associated with the release of high levels of active TGF-β. This TGF-β can further upregulate PTHrP expression by cancer cells, thereby forming a positive feedback loop called a “vicious cycle” [217].

Many specific studies have analyzed the role of TGF-β in lung and liver metastases. In a previous study, increased TGF-β levels were shown to lead to lung metastases in the MMTV/PyVmT transgenic model of metastatic breast cancer [218]. In addition, in breast cancer cells, TGF-β induced ANGPTL4 via the SMAD signaling pathway, and this cytokine could disrupt lung capillary walls and seed pulmonary metastases [219]. Because there are inherent differences in the microvasculature of these two tissues, lung metastasis requires robust extravasation functions provided by ANGPTl4 and other factors and additional lung colonizing functions achieved by ID1/ID3 [220]. Therefore, the vasculature disruptive mechanism provides a selective invasive advantage in the lung but not bone. Another study showed that the WNT signaling inhibitor Dickkopf 1 (DKK1) is a key factor for this metastatic preference; it reduces the recruitment of macrophages and neutrophils by the WNT/PCP-RAC1-JNK pathway and inhibits the level of tumor-derived TGF-β to inhibit lung metastasis. Thus, tumor cells that highly secrete DKK1 tend to metastasize to bone, while those with low DKK1 secretion tend to metastasize to the lung [221].

Statistical analysis of gene expression profiles revealed that TGF-β signaling is the most significant gene pathway in liver metastases of colorectal cancer [222]. Exosomes (small membrane vesicles with a size ranging from 40 to 100 nm) from pancreatic cancer induce Kupffer cells to release more TGF-β, which in turn activates the fibrotic pathway and forms a proinflammatory environment that supports pancreatic cancer metastasis [223].

### 6.2. TGF-β Promotes Angiogenesis

Regardless of the primary or secondary tumor, angiogenesis occurs once the tumor is more than 1–2 mm in diameter; therefore, a rich blood supply is necessary to provide nutrients and oxygen for tumor growth and metastasis [224]. Tumor cells secrete a variety of growth factors, including TGF-β, to accelerate the development of cancer by inducing angiogenesis (Figure 6B) [225].

On endothelial cells TGF-β can bind to the type I receptor, activin receptor-like kinase 1 (ALK-1), prompting downstream signaling involving intracellular and nuclear proteins (SMADs and Id1) and leading to a proangiogenic response [226]. Furthermore, TGF-β1 and hypoxia are potent inducers of vascular endothelial growth factor (VEGF) expression in tumor cells, and oncogenes, especially RAS, can also combine with the tumor microenvironment, providing the foundation for tumor cell invasion and angiogenesis [227]. Using a mouse mammary carcinoma model, researchers confirmed that VEGF expression in peri-necrotic areas is synergized by both hypoxia and TGF-β1, further showing that this cooperation is achieved through hypoxia-inducible factor (HIF)-1α physically associating with SMAD3 [228]. For example, TGF-β increases the expression of VEGF-C by coordinating with sine oculis homeobox homolog 1 (SIX1) in tumor cells, promoting tumor lymph angiogenesis and lymph node metastasis [229]. Likewise, TGF-β1 could promote macrophages to secrete more VEGF via the TβRII/SMAD3 signaling pathway in oral squamous cell carcinoma [230]. TGF-β and VEGF form a feedback loop through the Semaphorin3A/NEM axis; in general, abrogated VEGF inhibits the endothelial cell paracrine TGF-β1 and endothelial SMAD2/3 activation; in turn, TGF-β1 further stimulates endothelial Semaphorin3A expression [231]. Studies have shown that in RAS-transformed epithelial tumors, TGF-β significantly increases the expression of VEGF/VEGF-R, which has a powerful effect on capillary formation and migration of endothelial cells, thereby promoting angiogenesis in tumor cells [232]. TGF-β mediates the formation of new blood vessels by promoting connective tissue growth factor (CTGF) and angiogenic regulatory enzymes, such as matrix metalloproteinases (MMP-2, MMP-9, MMP-10, etc.) or by inhibiting tissue inhibitor of metalloproteinases (TIMP) [233].

### 6.3. TGF-β Promotes Immune Evasion

Under physiological conditions, the immune system is the most important element of human defense against cancer, and T lymphocytes and natural killer cells can recognize and specifically clear tumor cells [4,234]. However, tumor cells evade this immune surveillance through immune evasion of TGF-β, but the cellular mechanism by which TGF-β induces T cell dysfunction remains unclear. TGF-β may inhibit the proliferation of T cells and B cells and inhibit the production of immune factors by B lymphocytes (Figure 6C). In transgenic mouse studies, CD4+ and CD8+ T lymphocytes showed that expression of dominant-negative TβR2 was more effective in clearing thymoma and melanoma cells than in wild-type mice, which indicates that T lymphocytes are central targets for the negative regulation of TGF-β [235]. Interestingly, T cell production of TGF-β1 was shown to be a requirement for tumors to evade immune surveillance independent of TGF-β produced by tumors [236].

For example, regulation of tumor metastasis by TGF-β/SMAD signaling was found to be achieved by impairing the activity of tumor-infiltrating T cells [237,238], i.e., the infiltration level of CD3+, CD4+ and CD8+ cells and the proliferation and activity of T cells (such as the secretion of granzyme, FAS ligand (FASL), perforin and interferon (IFN)-γ) [239,240]. Regulatory T-cells (Tregs), whose excessive function inhibit antitumor immune responses, are another vital factor for TGF-β-mediated immune evasion by suppressing the proliferation and activation of CD8+ cytotoxic T-cells [241]. Effector Treg cells express high amounts of integrin αvβ8, which enables them to activate latent TGF-β, and tumor-derived TGF-β, in turn, induces FoxP3 expression and generates induced Treg cells [242]. Tregs can produce cell surface docking receptors for latent TGF-β, called glycoprotein A repetitions predominant (GARP). Further experiments revealed that overexpression of GARP leads to more TGF-β-releasing Treg cells and enhanced TGF-β signaling, tumor growth and metastasis in immunodeficient mice [243].

Additionally, TGF-β blocks immune surveillance by inhibiting migration and inducing apoptosis of antigen-presenting cells, such as dendritic cells (DCs), whose function is to mature and stimulate T cells during the immune response. Studies have shown that tumor-derived TGF-β significantly inhibits the proliferation of human CD4+ T cells activated by dendritic cells [244,245,246]. TGF-β also has an impact on myeloid cell functions. TGF-β in the tumor microenvironment polarizes tumor-associated macrophages (TAMs) from the pretumor (M2) phenotype to the antitumor (M1) phenotype. TGF-β inhibits neutrophil activity (i.e., degranulation) [247]. Furthermore, tumor-derived TGF-β polarizes the tumor-associated neutrophil (TAN) phenotype from N1 to N2 and induces a population of protumor TANs [248].

Moreover, the development of natural killer (NK) cells and T helper 1 (Th1) differentiation depend on TGF-β signaling. Functional studies have demonstrated that selective deletion of SMAD4 in NK cells impedes NK cell homeostasis and maturation, thereby reducing murine cytomegalovirus clearance [249]. A TGF-β-regulated transcription factor, T-bet, is responsible for Th1 differentiation and survival of activated CD4+ T cells via mediating CD122 expression and IL-15 signaling in Th1 cells [250].

## 7. Targeting the TGF-β Signaling Pathway in Cancer

Several TGF-β signaling inhibitors have been developed to curtail the aberrant TGF-β signaling characteristics of tumors. There are several types of TGF-β drugs in (pre)clinical development: ligand traps, antisense oligonucleotides (AONs), neutralizing antibodies, receptor domain-immunoglobulin fusions and receptor kinase inhibitors (Figure 7) [251]. While these agents have shown promise in the clinic, the complexity and pleiotropic nature of TGF-β tumor regulation render TGF-β targeted therapy a challenge. Related to this, careful administration/dosing of TGF-β therapies as well as judicious patient selection is needed to overcome on-target and off-target toxic side-effects [252].

Antisense oligonucleotides have been incorporated into immune cells to block translation or to degrade TGF-β mRNA. Belagenpumatucel-L (Lucanix) is a therapeutic vaccine for non-small cell lung cancer (NSCLC), which inhibits the expression of TGF-β2, thereby reducing the immunosuppressive effect of TGF-β2 and thus enhancing its antitumor effect [253]. Trabedersen (AP 12009) is an antisense oligonucleotide that specifically targets human TGF-β2 mRNA and has been used in the treatment of metastatic melanoma and pancreatic cancer [254]. Its clinical development has been put on hold (Clinical Trials: NCT00761280).

Multiple TβRI kinase inhibitors have been developed for the treatment of cancer and are under clinical trials. For example, a phase I study of Galunisertib (LY2157299), a TβR1 kinase inhibitor, showed an acceptable tolerability and safety profile in Japanese patients with advanced solid tumors [255] and is currently under clinical development in combination with immunotherapy, anti-PD-1 antibodies, including nivolumab and durvalumab (Clinical Trials: NCT02423343). SM16 (a TβRI kinase inhibitor) and 1D11 (a TGF-β neutralizing antibody) synergistically inhibited metastasis in combination with the agonistic OX40 antibody [256]. SD208, an inhibitor of TβR1, blocks the TGF-β/SMAD pathway, Matrigel invasion and expression of TGF-β target genes (PTHrP, IL-11, CTGF, and RUNX2, etc.) and was effective at preventing the development of bone metastases and decreasing the progression of established osteolytic lesions in melanoma and glioma models [257,258]. However, another study showed that SD-208 could not significantly reduce tumor growth and angiogenesis in SW-48 cells, a human colorectal cancer model, and thus, its efficiency still needs to be assessed [259].

Blockade of TGF-β ligand and receptor binding is a crucial mechanism for TGF-β inhibitory targeting agents. In a clinical phase I trial, PF-03446962, an anti-ALK1 antibody that displays (dose-dependent) antiangiogenic activity, was tested. This antibody was administered to patients who were resistant to anti-VEGF/VEGFR therapy [260]. However, in phase II, trials were stopped due to ineffectiveness (NCT01620970). Another TGF-β monoclonal antibody, Fresolimumab (GC-1008), was demonstrated to be safe and well tolerated in Phase I and Phase II trials [261]. Fresolimumab is an anti-TGF-β neutralizing antibody capable of neutralizing all human isoforms of TGF-β [262]. John et al. have shown that some melanoma or renal cancer patients with multiple doses of fresolimumab treatment experienced cutaneous lesions during phase 1 clinical trials [195]. Moreover, a phase I study noted that the maximum tolerated dose for the anti-TβRII monoclonal antibody LY3022859 could not be determined, but dose escalation beyond 25 mg was considered unsafe in patients with advanced solid tumors [263].

Finally, an interesting paper showed that a TGF-β inhibitor in combination with a PD-L1 inhibitor (Atezolizumab) may be able to remodel the matrix microenvironment and allow T cells to enter the interior of the tumor [237]. Furthermore, for chimeric antigen receptor (CAR) T cell therapies, CAR-T cells can reverse the immunosuppressive effect of TGF-β [264]. This finding suggests that combination therapies rather than mono-TGF-β therapies could prove to be the most effective approach in the future. 

## 8. Concluding Remarks

TGF-β acts as a tumor suppressor during the early phase of cancer progression and as a tumor promotor in advanced stages. The latter role, in particular, has been targeted as a potential therapeutic approach to inhibiting tumor growth and dissemination. However, developing effective treatments is severely hampered by the biphasic role of TGF-β in cancer and its function in many physiological processes, including the cardiovascular system. There has been some progress in the treatment of liver cancer and other types of cancer, but there is a pressing need for a greater understanding of pathological TGF-β mechanisms as well as greater clarity regarding protocols of therapy administration and patient selection.

With the success of immune checkpoint inhibitors for cancer therapy and considering the potent immune suppressive effects of TGF-β, it may be of particular interest to see whether TGF-β targeting agents can increase the efficiency and range of immune therapeutic agents. Such trials are underway, and the results are eagerly awaited. As highlighted in this review, other approaches could include the targeting of E3 ubiquitin ligases and deubiquitinating enzymes and miRNAs and lncRNAs, which control the stability of TGF-β receptors and SMAD proteins.

## Figures and Tables

**Figure 1 ijms-20-02767-f001:**
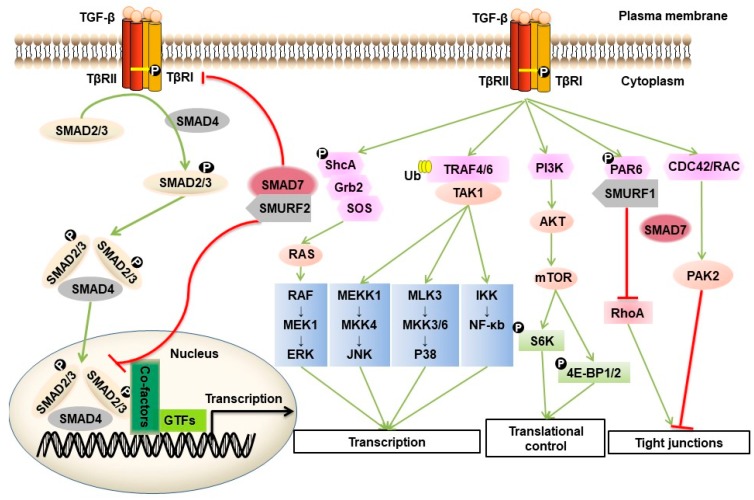
TGF-β/SMAD and non-SMAD signaling. TGF-β elicits its cellular responses by forming ligand-induced complex formation of TGF-β type I and type II cell surface receptors (i.e., TβRI and TβRII) that are endowed with serine/threonine kinase activity. The extracellular signal is transduced across the plasma membrane through the action of the constitutively active TβRII kinase that phosphorylates specific serine and threonine residues in the intracellular juxtamembrane GS domain of TβRI. Intracellular signaling is then initiated when the activated TβRI kinase phosphorylates or activates intracellular signal mediators. In the case of the canonical SMAD pathway, TβRI recruits and phosphorylates specific R-SMADs, e.g., SMAD2 and SMAD3, which can form heteromeric complexes with SMAD4. These transcription factor complexes then translocate into the nucleus and cooperate with other transcription regulators to regulate target gene expression. In the non-SMAD pathway, TGF-β receptors activate other pathways, including various branches of MAPK pathways, RHO-like GTPase signaling pathways and phosphatidylinositol-3-kinase (PI3K)/AKT pathways. Inhibitory signals are indicated with inhibitory red arrows; Stimulatory signals are indicated with green arrows.

**Figure 2 ijms-20-02767-f002:**
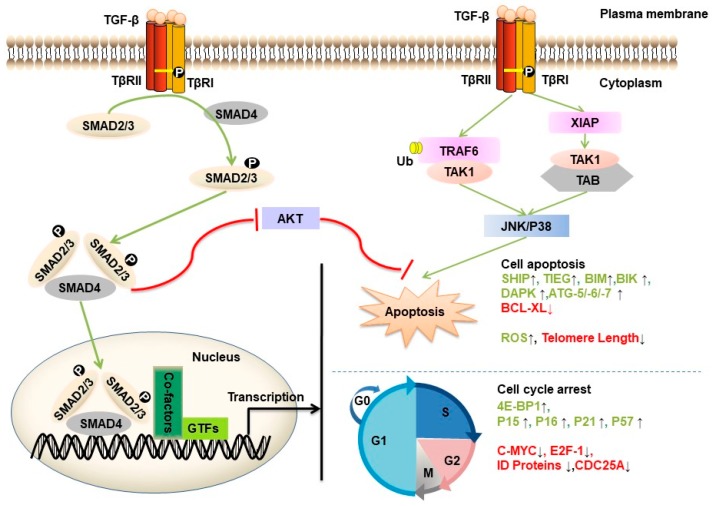
TGF-β-induced growth inhibition and apoptosis. The TGF-β/SMAD pathway can arrest cells in the G1 phase of the cell cycle by modulating the expression of specific genes, including induction of 4EBP1, p15INK4b, p16INK4A, p21WAF1 and p57KIP2 and repression of Id proteins, E2F, c-MYC and CDC25a genes. The TGF-β/SMAD pathway can induce cell apoptosis by inducing the expression of proapoptotic genes, such as BCL-XL, BIM, BIK, SHIP and TIEG. The TGF-β activation of TAK-1 occurs via TRAF6 and the adaptor XIAP-mediated TAB/TAK-1 complex and GADD45β. Activation of NF-κB by the TAK pathway stimulates p38/JNK phosphorylation, which has been reported to lead to apoptosis. Additionally, the TGF-β signaling pathway can be coupled to the cell death machinery through ROS, autophagy activation (ATG-5/-6/-7), induction of DAPK expression, epigenetic changes and shortening of telomere length (regulating hTERT). Inhibitory signals are indicated with inhibitory red arrows; Stimulatory signals are indicated with green arrows. SMAD-mediated transcriptional events are indicated with the black arrow.

**Figure 3 ijms-20-02767-f003:**
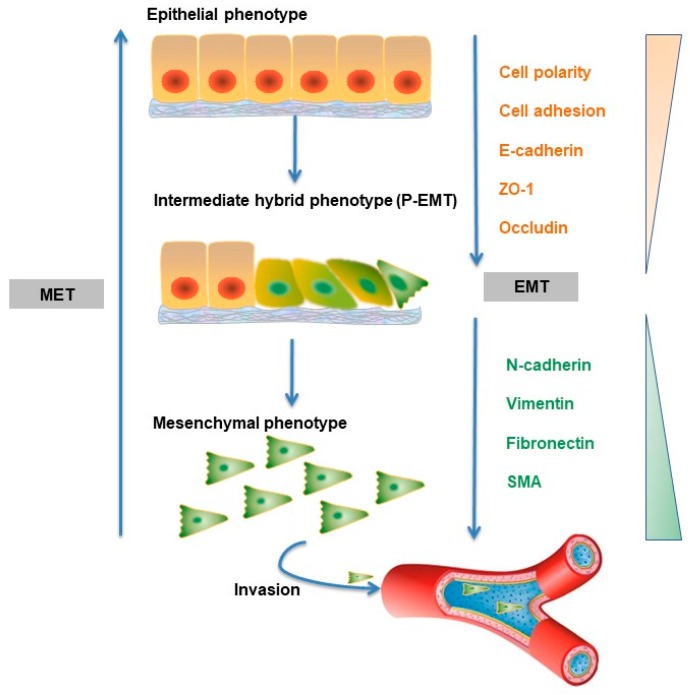
TGF-β mediates EMT. TGF-β is a strong promoter of EMT, which is characterized by a loss of epithelial and gain of mesenchymal markers. Polar epithelial cells remodel into highly migratory mesenchymal cells, followed by decreased adhesion of cells and loss of polarity and tight junctions. TGF-β via SMAD or non-SMAD signaling pathways can induce the expression of several EMT-TFs, such as SNAIL1, SNAIL2, ZEB1/2 and TWIST. The migratory and invasive mesenchymal-like tumor cells extravasate from primary lesions into blood or lymphatic vessels and then intravasate to distant sites via, where they form metastatic colonies upon MET.

**Figure 4 ijms-20-02767-f004:**
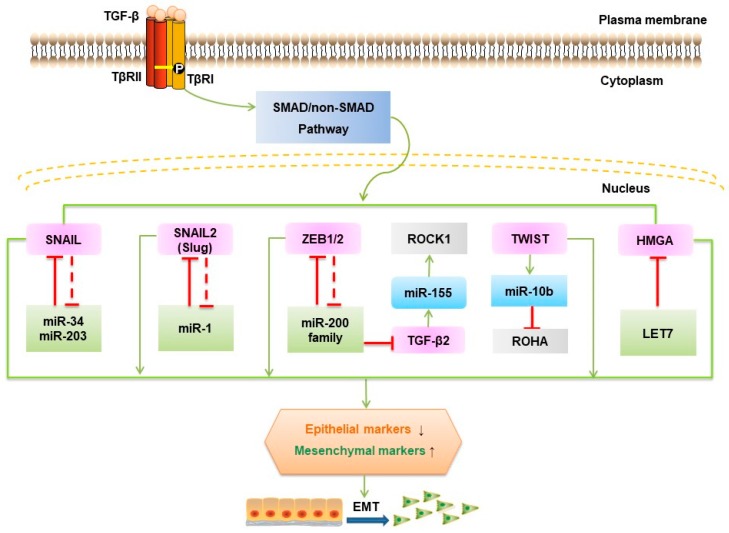
MicroRNAs in TGF-β-induced EMT. At the heart of TGF-β-induced EMT, there are two main double-negative feedback regulatory loops of miRNAs, e.g., the SNAIL1/miR-34 family and ZEB/miR-200 family and the autocrine TGF-β/miR-200 negative feedback loop. Specifically, TGF-β downregulates miR-200 family members, thereby increasing ZEB1 and ZEB2 mRNA levels indirectly, and ZEB binds to promoters of the miR-200 members to repress their expression, thus constituting a double-negative regulatory loop. The same situation occurs in SNAIL1 and miR-34, which are directly linked to p53 status. For the autocrine TGF-β/miR-200 system, autocrine TGF-β positively regulates the expression of SNAIL1 and then increases ZEB mRNA and protein levels, further downregulating miR-200. Inhibitory signals are indicated with inhibitory (dashed) red arrows; Stimulatory signals are indicated with green arrows.

**Figure 5 ijms-20-02767-f005:**
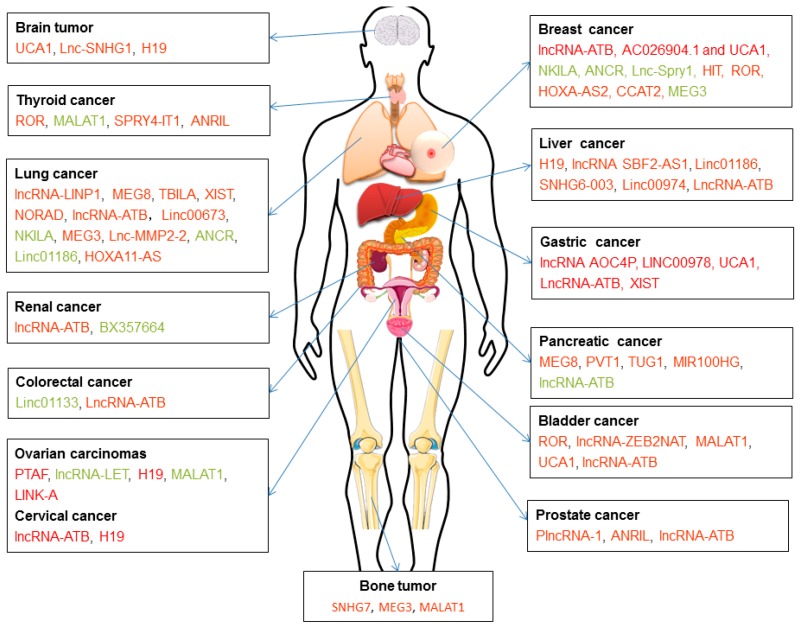
LncRNAs in TGF-β signaling and TGF-β-induced EMT. LncRNAs associated with TGF-β signaling and TGF-β-induced EMT in various cancer cell types. LncRNAs with high expression in tumor tissues (in red) and low expression in tumor tissues (in green).

**Figure 6 ijms-20-02767-f006:**
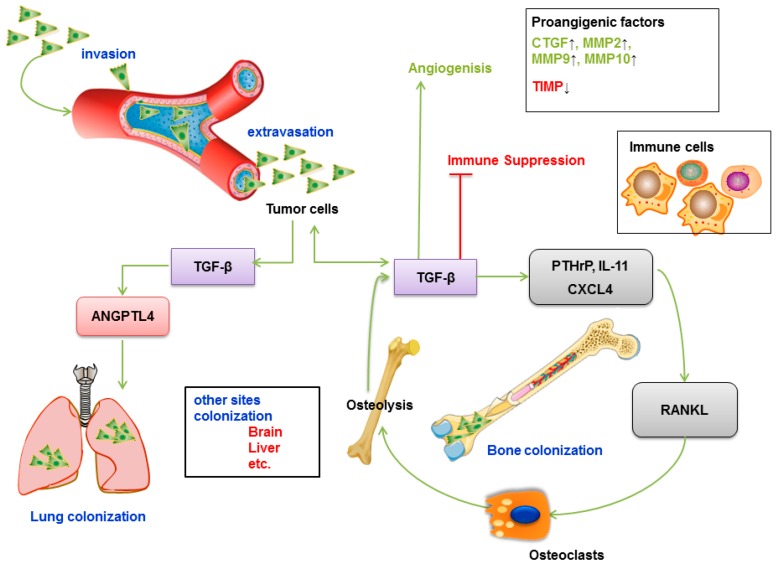
TGF-β mediates metastasis. TGF-β produced by cancer cells can alter the bone microenvironment by inducing the expression of osteolytic factors like PTHrP and IL11. The osteoclast bone resorption via RANKL production by osteoblasts results in more release of TGF-β, which in turn acts on tumor cells thereby creating a positive feedback loop called a “vicious cycle”. Chemokine receptor CXCL4 and angiogenesis inducer connective tissue growth factor (CTGF) are also key modulators induced by TGF-β in this process. Another important factor of this metastatic preference is the Wnt antagonist Dickkopf 1 (DKK1); cells that highly secrete DKK1 tend to metastasize to bone, while low DKK1-secreting tumor cells tend to metastasize to the lung. TGF-β leads to transcriptional upregulation of proangiogenic factors, including CTGF, matrix metalloprotease (MMP)2, MMP-9, and MMP10, or inhibition of TIMP to mediate the formation of new blood vessels. TGF-β inhibits the proliferation of T cells and B cells and inhibits the production of immune factors by B lymphocytes. In addition, TGF-β-induced angiopoietin-like 4 (ANGPTL4) via the SMAD signaling pathway is proangiogenic and can disrupt lung capillary walls and seed pulmonary metastases. Inhibitory signals are indicated with inhibitory red arrows; Stimulatory signals are indicated with green arrows.

**Figure 7 ijms-20-02767-f007:**
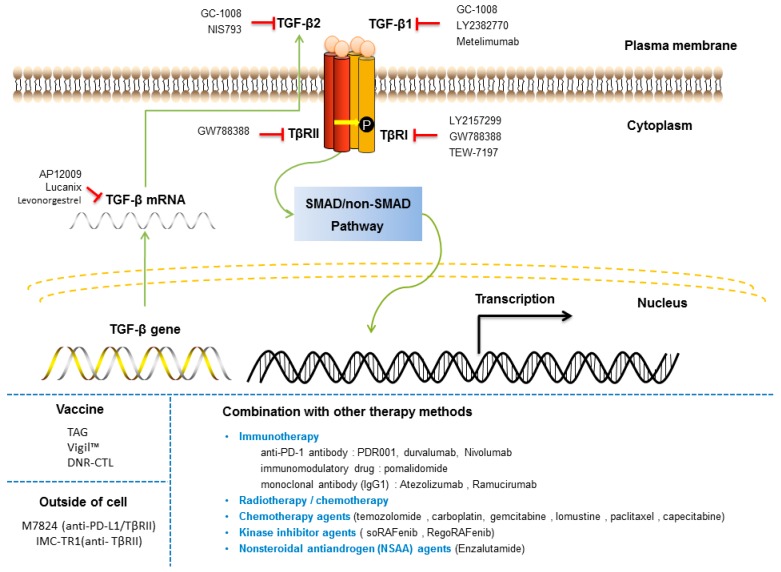
Targeting the TGF-β signaling pathway in cancer. Various molecular mechanisms by which TGF-β signaling is targeted for therapeutic gain are depicted. TGF-β inhibitory targeting agents (Table 1), including TβRI kinase inhibitors, AON targeting ligand or receptor gene expression, and antibodies interfering with ligand-receptor interactions, have been developed to curtail excessive TGF-β pathway activation. Inhibitory signals are indicated with inhibitory red arrows; Stimulatory signals are indicated with green arrows.

**Table 1 ijms-20-02767-t001:** LncRNAs involved in TGF-β signaling, focusing on those that impact TGF-β-induced EMT. LncRNAs with high expression in tumor tissues (in red) and low expression in tumor tissues (in green).

Cancer Type	LncRNA	Function and Mechanism of Action	Example of Key Findings or Experiments
Breast cancer	lncRNA-ATB [144]	Functions as a sponge of miR-141-3p, increasing ZEB1 and ZEB2 expression.	Knockdown results in a morphological change of breast cancer cells from spindle-like to round shape and in a remarkable inhibition of cell migration and invasion.
	AC026904.1 and UCA1 [145]	Functions as an enhancer RNA (eRNA) to activate Slug gene transcription in the nucleus, whereas UCA1 exerts a competitive endogenous RNA (ceRNA) for titrating miR-1 and miR-203a to promote Slug expression at the post-transcriptional level in the cytoplasm.	Knockdown of either AC026904.1 or UCA1 prolongs survival time of the nude mice bearing D3H2LN mammary tumors, these two genes exert critical roles in TGF-β-induced EMT and promote invasion in metastatic breast cancer.
	NKILA [146]	Suppresses TGF-β-induced EMT by blocking NF-κB signaling	Overexpression reduces TGF-β-induced tumor metastasis in vivo.
	ANCR [147]	Functions as a downstream effector molecule, down-regulated by TGF-β1, and is essential for TGF-β1-induced EMT by decreasing RUNX2 expression.	Ectopic expression partly attenuates the TGF-β1-induced EMT and knockdown promotes TGF-β1-induced EMT and metastasis in breast cancer.
	Lnc-Spry1 [141]	Functions as an immediate-early regulator of EMT that is downregulated by TGF-β, affecting the expression of TGF-β-regulated gene targets; alternative splicing by U2AF65 splicing factor; isoform switching of fibroblast growth factor receptors.	Knockdown promotes a mesenchymal-like phenotype and results in increased cell migration and invasion.
	lncRNA-HIT [148]	Ectopic expression disrupts tight junction by targeting E-cadherin.	Knockdown results in decrease of cell migration, invasion, tumor growth, and metastasis.
	linc-ROR [149,150]	Functions as a sponge of miR-145 and therefore upregulate the expression of ARF6, which regulates adhesion and invasion properties of breast tumor cells through E-cadherin.	Regulates the cancer stem cell phenotype in Triple-negative Breast Cancer, which plays a critical role in drug resistance and metastasis.
	HOXA-AS2 [151]	Functions as an endogenous sponge of miR-520c-3p, and controls the expression of miR-520c-3p target genes, TβR2 and RELA, in breast cancer cells.	Knockdown inhibits the progression of breast cancer cells in vitro and in vivo.
	CCAT2 [152]	Knockdown causes cell cycle arrested in G0/G1 phase, promotes cell apoptosis and downregulates the protein expression levels of TGF-β, Smad2 and α-SMA in breast cancer cells.	Down-regulation inhibits the proliferation, invasion and migration in breast cancer cells.
	MEG3 [134,153,154]	Regulates the TGF-β pathway genes through formation of RNA-DNA triplex structures, and downregulates AKT and functions as a sponge of miR-421 to regulate E-cadherin expression.	Ectopic expression inhibits in vivo tumorigenesis and angiogenesis in a nude mouse xenograft model.
Gastric cancer	LINC00978 (known as MIR4435-2HG and AK001796) [155]	Knockdown inhibits the activation of TGF-β/SMAD signaling pathway and EMT in GC cells.	Knockdown inhibits the proliferation, migration and invasion and decreases the in vivo tumorigenicity of GC cells in mice.
	UCA1 [156]	Knockdown inhibits TGF-β1-induced-EMT process and the effect could be partly restored by TGF-β1 treatment.	A potential oncogenic factor by regulating GC cells proliferation, invasion, and metastasis under TGF-β1 induction.
	lncRNA-ATB [157,158,159]	Induced by TGF-β and functions as a ceRNA of miR-141-3p or miR-200s.	A novel biomarker of lncRNA, correlated with increased invasion depth, more distant metastasis and advanced tumor-node-metastasis stage.
	XIST [160]	Functions as a competing endogenous lncRNA (ceRNA) to regulate TGF-β1 by sponging miR-185 in GC.	sh-XIST inhibited GC development in vitro.
Ovarian carcinomas	LncRNA-LET [161]	Regulates EMT process and the expression of TIMP2 and activates the Wnt/β-catenin and Notch signaling pathways.	Overexpression inhibits cell viability, migration and EMT process, and increases apoptosis in KGN cells
	H19 [162]	Functions by competing with miR-370-3p to regulate TGF-β-induced EMT in ovarian cancer.	Knockdown suppresses TGF-β-induced EMT, while H19 overexpression promotes TGF-β-induced EMT.
	MALAT1 [163,164]	TGF-β increases its expression by inhibiting miR-200c; MALAT1 interacts with MARCH7, which regulates TGF-β-smad2/3 pathway by interacting with TβR2, via miR-200a as a ceRNA.	Interrupts the interaction between miR-200c/MALAT1 decreases the invasive capacity of EEC cells and EMT in vitro and inhibits EEC growth and EMT-associated protein expression in vivo; This LncRNA plays an important role in TβR2-Smad2/3-MALAT1/MARCH7/ATG7 feedback loop mediated autophagy, migration and invasion in ovarian cancer.
	LINK-A [165] cervical cancer	TGF-β1 treatment has no effects on LINK-A expression, and there is no clear mechanism.	Overexpression increases expression of TGF-β1 in ovarian carcinoma cells and promotes cell migration and invasion and this effect can be attenuated by TGF-β1; Plasma levels are correlated with distant tumor metastasis but not tumor size.
	lncRNA-ATB [166]	No clear mechanism.	A promising prognostic marker that correlates with the malignant phenotypes and poor prognosis of cervical cancer
	PTAF [167]	Functions by competing with miR-25 and affecting SNAI2 to regulate the expression of many EMT-related protein-coding genes in OvCa.	A mediator of TGF-β signaling. Upregulation induces elevated SNAI2, which in turn promoted OvCa cell EMT and invasion; knockdown inhibits tumor progression and metastasis in an orthotopic mouse model of OvCa.
Bladder cancer	lncRNA-ZEB2NAT [168]	Induced by TGF-β, and can regulate EMT process by affecting ZEB2 protein level.	Knockdown reverses CAF-CM-induced EMT and invasion of cancer cells, as well as reduces the ZEB2 protein level.
	MALAT1 [169]	Induced by TGF-β and regulates EMT by negatively correlated E-cadherin, N-cadherin and fibronectin expression by zeste 12 (suz12) in vitro and in vivo.	Overexpression is significantly correlated with poor survival in patients with bladder cancer. Inhibition of malat1 or suz12 suppresses the migratory and invasive properties induced by TGF-β and inhibits tumor metastasis in animal models.
	UCA1 [170]	Induced by BMP9 through phosphorylated AKT and there are no clear mechanism.	Its BMP-9-induced expression associates with increased proliferation and migration of bladder cancer cells. The promoting effect of BMP9 is rescued after interfering with UCA1 in BMP9 overexpressed bladder cancer cells both in vitro and in vivo.
	lncRNA-ATB [171]	Is upregulated by TGF-β, acting as a molecular sponge of miR-126 and regulate the direct target of miR-126 (KRAS).	Its overexpression significantly promotes cell viability, migration, and invasion in T24 cells.
	PlncRNA-1 [172]	Regulates the cell cycle, cyclin-D1 and EMT in prostate cancer cells through the TGF-β1 pathway.	Functions as an oncogene.
	ROR [173]	Knockdown can reverse TGF-β1-induced-EMT phenotype in SGC-996 and Noz cells. However, there are no clear mechanism.	High expression is associated with poor prognosis in gallbladder cancer patients and knockdown inhibits cell proliferation, migration, and invasion.
Prostate cancer	ANRIL [174]	Regulates let-7a/TGF-β1/Smad signaling pathway.	Overexpression promotes the proliferation and migration of prostate cancer cells.
	lncRNA-ATB [175]	Upregulated by TGF-β, stimulates EMT associated with ZEB1 and ZNF217 expression levels via ERK and PI3K/AKT signaling pathways.	Overexpression promotes, and knockdown of lncRNA-ATB inhibits the growth of prostate cancer cells via regulations of cell cycle regulatory protein expression levels.
Brain tumor	lnc-SNHG1 [176]	Activates the TGFBR2/SMAD3 and RAB11A/Wnt/β-catenin pathways in pituitary tumor cells via sponging miR-302/372/373/520.	Promotes the progression of pituitary tumors, ectopic expression of lnc-SNHG1 promotes cell proliferation, migration, and invasion, as well as the EMT, by affecting the cell cycle and cell apoptosis in vitro and tumor growth in vivo.
	UCA1 [177]	Functions as a ceRNA of miR-1 and miR-203a to promote Slug expression, which underlies TGF-β-induced EMT and stemness of glioma cells.	Knockdown attenuates EMT and stemness processes and their enhancement by TGF-β.
Lung cancer	lncRNA-LINP1 [178]	TGF-β1 inhibits its transcription in a SMAD4-dependent manner.	Inhibits TGF-β-induced EMT and thereby controlling cancer cell migration, invasion, and stemless in lung cancer cells.
	TBILA [179]	Induced by TGF-β and functions via cis-regulating HGAL and activating S100A7/JAB1 signaling.	Promotes non-small cell lung cancer progression in vitro and in vivo.
	XIST [180,181]	Functions as an endogenous sponge by directly binding to miR-137, negatively regulating its expression and regulating Notch gene expression.	Overexpression inhibits proliferation and TGF-β1-induced EMT in A549 and H1299 cells, regulating proliferation and TGF-β1-induced EMT in NSCLC, which could be involved in NSCLC progression.
	NORAD [182]	Affects the physical interaction of its binding partner (importin β1) with Smad3, and then inhibits the nuclear accumulation of Smad complexes in response to TGF-β.	Stimulates TGF-β signaling and regulates TGF-β-induced EMT-like phenotype in A549 cells.
	lncRNA-ATB [183]	Regulates EMT by down-regulating miR-494 in A549 cells, which in turn increases phosphorylated levels of AKT, JAK1, and STAT3.	Overexpression promotes proliferation, migration, and invasion of A549 cells. In contract, ATB silence shows the opposite influence.
	linc00673 [184]	Functions as a sponge of miR-150-5p and indirectly modulates the expression of key EMT regulator ZEB1.	Inhibition attenuates the tumorigenesis ability of A549 cells in vivo.
	NKILA [185]	Expression is regulated by TGF-β and regulates EMT process by inhibiting the phosphorylation of IκBα and NF-κB activation to attenuate Snail expression.	Inhibits migration, invasion and viability of NSCLC cells. Lower NKILA expression are correlated with lymph node metastasis and advanced TNM stage.
	MEG3 [186]	Associates with JARID2 and the regulatory regions of target genes to recruit the complex by epigenetic regulation (PRC2/JARID2/ H3K27 axis).	Knockdown inhibits TGF-β-mediated changes in cell morphology and cell motility characteristic of EMT and counteracts TGF-β-dependent changes in the expression of EMT-related genes; In contrast, overexpression enhances these effects.
	lnc-MMP2-2 [187]	ls highly enriched in TGF-β-mediated exosomes and might function by increasing the expression of MMP2 through its enhancer activity.	Knockdown affects lung cancer invasion and vascular permeability.
	ANCR [188]	Inhibits NSCLC cell migration and invasion by downregulating TGF-β1 expression, however TGF-β1 treatment shows no significant effects on ANCR expression but promotes NSCLC cell migration and invasion.	Low expression level indicates shorter postoperative survival time of NSCLC patients, whereas, ectopic expression inhibits NSCLC cell migration, invasion and downregulated TGF-β1 expression, and this effect can be attenuated by TGF- β1.
	LINC01186 [189]	Functions as a mediator of TGF-β signaling, is down-regulated by TGF-β1 regulating EMT by Smad3	Knockdown promotes cell migration and invasion, whereas, overexpression prevents cell metastasis.
	HOXA11-AS [190]	Regulates the expression of various pathways and genes, especially DOCK8 and TGF-β pathway, however, there is no clear mechanism.	Its expression may determine the overall survival and disease-free survival of lung adenocarcinoma patients in TCGA.
Liver cancer	lncRNA SBF2-AS1 [191]	Functions as a ceRNA of miR-140-5p and regulates the expression of TβR1.	Knockdown inhibits the proliferation, migration and invasion of HCC cells and attenuate the development of HCC tumor in vivo.
	SNHG6-003 [166]	Functions as a ceRNA of miR-26a/b and thereby modulates the expression of transforming growth factor-β-activated kinase 1 (TAK1).	Ectopic expression in HCC cells promotes cell proliferation and induces drug resistance, whereas knockdown promotes apoptosis. High expression of SNHG6-003 closely correlated with tumor progression and shorter survival in HCC patients.
	LINC00974 [192]	Interacts with KRT19, as a sponge of miR-642, activating the Notch and TGF-β pathways.	Knockdown inhibits cell proliferation and invasion with an activation of apoptosis and cell cycle arrest both in vitro and in vitro.
	lncRNA-ATB [132]	Upregulated by TGF-b and can induce EMT and invasion by acting as ceRNA of miR-200 family and increasing ZEB1/2; promotes organ colonization by binding IL-11 mRNA, autocrine induction of IL-11, and triggering STAT3 signaling.	Promotes the invasion-metastasis cascade in hepatocellular carcinoma.
Pancreatic cancer	PVT1 [193]	Regulates EMT process via TGF-β1/Smad signaling.	Acts as an oncogene in pancreatic cancer, knockdown of PVT1 inhibits viability, adhesion, migration and invasion.
	MEG8 [194]	MEG8, which shares the DLK1-DIO3 locus with MEG3, can induce the recruitment of EZH2 protein to miR-34a and miR-203 genes for histone H3 methylation and transcriptional repression, inducing EMT-related cell morphological changes and increases cell motility in the absence of TGF-β by activating the gene expression program required for EMT.	Plays critical role in TGF-β-induced EMT in A549 lung cancer and Panc1 pancreatic cancer cells.
	TUG1 [195]	Regulates EMT process though TGF-β/Smad pathway	Overexpression increases cell proliferation and migration capacities, enhancing the proliferation and migration of pancreatic cancer cells
	MIR100HG [196]	Induced by TGF-β, and this gene contains miR-100, miR-125b and let-7a in its intron, through SMAD2/3. These miRNAs regulate a multitude of genes involved in the inhibition of p53 and DNA damage response pathways.	Plays prominent role in metastasis of pancreatic cancer.
	lncRNA-ATB [197]	No clear mechanism.	Low expression levels are correlated with lymph node metastases neural invasion, and clinical stage and worse overall survival prognoses of patients.
Renal cancer	BX357664 [197,198]	Blocks the TGF-β1/p38/HSP27 pathway.	Upregulation reduces migration, invasion, and proliferation capabilities in RCC cells.
	lncRNA-ATB [199]	No clear mechanism.	Its expression is correlated with metastases and promotes cell migration and invasion in renal cell carcinoma. Knockdown could inhibit cell proliferation, trigger apoptosis, reduce epithelial-to-mesenchymal transition program and suppress cell migration and invasion.
Colorectal cancer	lncRNA-ATB [199,200,201]	Upregulated by TGF-β, and suppresses E-cadherin expression and promoting EMT process.	High expression is significantly associated with greater tumor size, depth of tumor invasion, lymphatic invasion, vascular invasion, and lymph node metastasis.
	LINC01133 [202]	Downregulated by TGF-β and its expression is positively correlated with E-cadherin, and negatively correlated with Vimentin. Directly interacting with SRSF6, which promotes EMT and metastasis in CRC cells.	Inhibits EMT and metastasis in colorectal cancer (CRC) cells and low LIINC01133 expression in tumors with poor survival in CRC samples.
Bone tumor	MEG3 [203]	Represses Notch and TGF-β signaling pathway by inhibiting Notch1, Hes1, TGF-β and N-cadherin expression, and increasing E-cadherin.	Overexpression represses cell proliferation and migration ability.
	MALAT1 [204]	Induced by TGF-β and its overexpression decreases E-cadherin level, however, this effect was partially reversed by EZH2 knockdown.	Overexpression promotes cell metastasis, a potential diagnostic and prognostic factor in osteosarcoma.
	SNHG7 [205]	Inhibits tumor suppressor miR-34s signals and the targets of miR-34a, including proliferation-related Notch1, apoptosis-related BCL-2, cell cycle-related CDK6, and EMT-related SMAD4.	Knockdown delays the tumor growth in osteosarcoma tissues.
Thyroid cancer	MALAT1 [206]	Induced by TGF-β and supports a role for MALAT1 in EMT in thyroid tumors.	Functions as an oncogene and as a tumor suppressor in different types of thyroid tumors.
	ROR [207]	Functions as a competing endogenous RNA (ceRNA) of sponging miR-145.	Expression is induced by TGF-β in cells undergoing EMT.
	SPRY4-IT1 [208]	Knockdown increases the levels of TGF-β1 and p-Smad2/3 and this effect could be rescued by the interference of TGF-β1.	A novel prognostic factor, which correlates with poor prognosis and exhibits that silenced SPRY4-IT1 inhibited the proliferative and migratory abilities of TC cells.
	ANRIL [209]	Reduces p15INK4B expression through inhibiting TGF-β/Smad signaling pathway.	Promotes invasion and metastasis of TC cells, and the silencing of ANRIL inhibits the invasion and metastasis of TPC-1 cells.

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
