# Peer review of "TGF-β-Mediated Epithelial-Mesenchymal Transition and Cancer Metastasis"

_ijms, 2019, doi:10.3390/ijms20112767_

Round 1
Reviewer 1 Report
Although this review contains important information and comprehensively described the multiple relationship of TGFb to other tumour suppressor genes, microRNA, non-coding RNA and others. Before this review can be accepted, extensive English editing is required. There are multiple Grammatic error, including: spacing, missing word table with red colour with no indicative meaning.
Author Response
We had the manuscript checked by Nature Editing service. Also we have gone through the manuscript carefully to correct spelling, spacing, missing word table with red colour with no indicative meaning mistakes.
See for file for certificate.

Reviewer 2 Report
This is an excellent and comprehensive review that focuses on a controversial aspect of cancer cell biology. The paper provides a nicely-constructed discussion of the differing views of the EMT process while providing an important reference database that investigators new to the field will find invaluable. The description of the intersecting pathways fundamental to attaining the various phases of the plastic transition are well described without being overly complicated. There are a number of missing punctuation marks and abbreviations that are need addition of the appropriate greek symbols. This is a minor point but one that requires attention to facilitate the readability of the text. Careful attention to proof reading should make a revision relatively easy.
Author Response
We had the manuscript checked by Nature Editing service. We also checked again the manuscript for spelling errors and clarified text where needed.
See for file for certificate.
